# Coexistence between Humans and ‘Misunderstood’ Domestic Cats in the Anthropocene: Exploring Behavioural Plasticity as a Gatekeeper of Evolution

**DOI:** 10.3390/ani12131717

**Published:** 2022-07-02

**Authors:** Eugenia Natoli, Carla Litchfield, Dominique Pontier

**Affiliations:** 1Canile Sovrazonale, ASL Roma 3 (Local Health Unit Rome 3), 00148 Rome, Italy; 2Conservation Psychology and Applied Animal Behaviour Research Group, Justice and Society (Discipline of Psychology), University of South Australia, Adelaide, SA 5000, Australia; carla.litchfield@unisa.edu.au; 3Laboratoire de Biométrie et de Biologie Évolutive, UMR CNRS5558, Université C. Bernard Lyon 1, 43 Bd du 11 nov. 1018, 69622 Villeurbanne, France; dominique.pontier@univ-lyon1.fr

**Keywords:** domestic cat, Anthropocene, behavioural plasticity, new trait evolution

## Abstract

**Simple Summary:**

The free-ranging unowned domestic cat (unowned—not under human control with respect to movement and sexual behaviour), living in the Anthropocene, can live a strictly solitary life or in socially structured groups, depending on environmental conditions. This paper explores the evidence for evolution of new traits (behavioural, morphological, physiological, immunological) in domestic cats, to adapt to the variety of ecosystems they now successfully inhabit. While the domestic cat ancestor lived a strictly solitary life, unowned free-ranging cats today may live in multi-male/multi-female colonies in urban city centres, where they are dependent on food provided by people. Urban free-ranging cats are now more social, which has been reflected in different breeding patterns, lower infanticide, more frequent affiliative interactions in general, and different spatial groupings. This means there is a potential for domestic cat behaviour to be ‘misunderstood’. Recognising that negative impacts of free-ranging domestic cats in urban fringe areas must be mitigated, we discuss how understanding behavioural plasticity and other recently evolved traits of domestic cats may lead to management strategies that maximise health and welfare of cats, wildlife, and humans.

**Abstract:**

Welfare and management decisions for unowned free-ranging cats in urban environments should no longer be based on knowledge about behavioural ecology of solitary cats living and breeding in more natural ‘wild’ environments. We provide evidence that urban free-ranging domestic cats in the Anthropocene have responded to rapidly changing environments, such as abundance of food and higher population densities of conspecifics by adapting their behaviour (behavioural plasticity—the ability of a genotype (individual) to express different behaviours according to its environment) and social organisation to living in complex social groups, especially those living in colonies. Urban free-ranging cats are now more social, as demonstrated by different breeding patterns, lower infanticide, more frequent affiliative interactions in general, and different spatial groupings. We argue that this knowledge should be disseminated widely, and inform future research and strategies used to manage free-ranging cats across environments. Understanding behavioural plasticity and other recently evolved traits of domestic cats may lead to management strategies that maximise health and welfare of cats, wildlife, and humans—otherwise domestic cat behaviour may be ‘misunderstood’. Importantly, interdisciplinary research using expertise from biological and social sciences, and engaging human communities, should evaluate these management strategies to ensure they maintain optimal welfare of free-ranging domestic cats while preserving biodiversity and protecting wildcats.

## 1. Density of Unowned Domestic Cats

The domestic cat (*Felis s. catus* L.) is now present on all continents, with the exception of the poles and certain islands. It lives in a wide variety of environments where the spatial and temporal distribution of resources vary widely, so that the species includes individuals with very different lifestyles (reviewed in [1], this Special Issue). This ubiquitous nature of the domestic cat is a sign of high behavioural adaptability [2], as will be seen below. It is now generally accepted that behavioural studies of domestic cats, and other opportunistic animal species (opportunistic species—a species that can quickly exploit new resources as they arise, for example, by rapidly colonising a new environment [3]) (Norway rats *Rattus norvegicus*, *Berkenhout 1769*, [4]; house mice *Mus musculus* L., [5]; red foxes *Vulpes vulpes* L., [6]; starlings *Sturnus vulgaris* L., [7]; yellow-legged gulls, *Larus michahellis*, *Naumann 1840*, [8]), have shown that the most important environmental variable in determining the density and spatial distribution of the animal population is the quantity and distribution of trophic resources [9]. But substantial variation in the density of the descendant species, compared to the ancestor, can affect all aspects of life for these animals, from spacing patterns to social organisation to the mating system, from the dynamics of prey–predator interaction to the dynamics of parasite–host interaction. Among domesticated species, the domestic cat represents a highly valuable and unique model to study these phenomena because it can live a strictly solitary life or in socially structured groups, depending on environmental conditions. This allows comparison of population characteristics at the intra-specific level to identify factors that predispose species to evolve sociality, a privilege for students of behaviour that few mammalian species allow.

Studies carried out in contrasting ecological conditions have shown that in the sub-Antarctic island environment, the unowned domestic cats, which can be defined as ‘feral cats’ (feral cats—domestic cats that, after being domesticated, returned to a feral situation and are not dependent on humans for trophic resources.), live at a very low density (less than a cat/km^2^, [10]) in individual territories (Figure 1).

They therefore live a solitary lifestyle, very similar to that of the European wildcat (*Felis s. silvestris* Schreber, 1777) [11].

At the other extreme are the urban unowned free-ranging domestic cats (FRCs), living in colonies in the urban environment at a very high density (more than a thousand individuals per km^2^) and within a social spatial organisation [12,13,14,15,16,17] (Figure 2).

The rural cat lives in an intermediate situation, typically within a human dwelling or property, with a relatively low density (about 250 individuals per km^2^, [13]) in small groups.

## 2. Spacing Pattern of Unowned Domestic Cats

Home range and territory should not be taken as synonyms; the former indicates the area of activity for an animal, patrolled but not necessarily defended against trespassing conspecifics. In contrast, the territory is an exclusive area actively defended by the resident cat (or by a group of resident animals), smaller than the home range and enclosed within it [15]. There are few studies in which this distinction is made, and, to our knowledge, they are in the urban context. In fact, in most cases, the research focused only on the home range, since it is easier to assess the area in which a cat’s activities occur (i.e., home range) than the actively defended part of this area (i.e., territory). As Liberg and colleagues emphasise [13], the utilisation of the term ‘territoriality’ requires active defence of the range (p. 133); of course, ‘There is a large asymmetry between the data needed to show range overlap and exclusive range’ [13] (p. 133). Intuitively, considering the small size of the spaces colonised by FRCs in the urban environment compared to, for example, the rural environment, it is easier to notice the difference between the home range and the territory of a resident male in the urban cat colonies. In every study that the first author (E.N.) has conducted on cat colonies in an urban environment, she has noticed that cats belonging to different neighbouring social groups tolerated each other, although they threatened each other in the overlapping marginal zones (one can assume, the marginal parts of the home range). But if an unfamiliar cat intruded into the innermost part of the group’s territory, it was attacked by the adult individuals (males and females) of the group ([15] p. 301), and the intruder aroused the curiosity/fear of the young. These behavioural observations apply to unneutered domestic cats.

Therefore, in this review, we use the term that the author(s) use for each of the studies we cite, which may be either ‘home range’ or ‘territory’.

The domestic cat is a territorial animal, whatever environment it lives in. This means that, when it lives as a solitary animal, it conducts its life within a large area containing resources such as food, shelters, and sexual partners, which the cat defends from the intrusion of conspecifics. Therefore, if sufficiently competitive, an adult male can occupy an area in an exclusive way, with the size of the area determined by the richness of prey species. The more resources available, the smaller the area and the lower the costs of controlling it. The hypothesised original spacing pattern is one in which the home range of an adult male includes two or three smaller home ranges of adult females [18].

In general, tomcats have a significantly larger home ranges than females (but see [19]), regardless of habitat and type of resources—natural prey and/or food distributed by humans, as also reported for the African wildcat [18].

In sub-Antarctic islands, individuals of both sexes are solitary, living in large non-overlapping home ranges [20,21,22], although there are some indications that, in Kerguelen islands, male home ranges can overlap in space, but not in time [23].

The spatial distribution pattern of rural unowned domestic cats resembles the pattern of African and European wildcats [18,24,25], consisting of a male owning and monitoring a relatively large home range, which includes the home ranges of two or three females. In this environment, males and females are solitary but females may also form small groups of closely related individuals associated with human dwellings [13,26,27].

In contrast, in the urban environment, when unowned domestic cats live socially, the territory belongs to the group and is defended by all adult male and female members of the group [15]. At the centre of the territory is the core area, the area of most intense and regular use, which represents the most important place in the life of a social group of mammals, where mostly high-ranking adult and young cats live.

## 3. The Social Organization of Unowned Domestic Cats

In environments where unowned free-ranging cats live as solitary animals—depending exclusively on prey for their survival—there is no social organisation and therefore no point in using the term ‘dominant’ male; that is, the concept of dominance hierarchy is only relevant to a social group [28,29]. The same is true for the rural environment as described in the previous section.

In urban ecosystems where cats depend on food provided by cat caretakers [30], cats organise themselves around feeding sites, forming highly structured, stable multi-male/multi-female groups called colonies [31]. Male and female home ranges and territories overlap [32,33]. Cats are fed by people who distribute the food in specific locations where all cats eat at the same time. In a rural environment, each cat (or several cats attached to the same farm) has its own feeding location provided by humans (26), i.e., different to that of the cat(s) living on the neighbouring farm. In contrast, in an urban environment, all cats, even dozens of cats, belonging to the same group share the same feeding location. This provisioning may have a negative impact (e.g., spread of viruses [34]), but also highlights the resilience of domestic cats who can adapt and survive in seemingly hostile environments. Provisioned cats live in areas around hospitals, in public gardens, and in historical ruins, as these places become favourable islands with shelters, surrounded by roads with heavy traffic. In a study conducted in a hospital in the city of Lyon (France), the home ranges of adult females were organised around a single feeding site. Adult males’ home ranges included up to three of five permanent feeding sites [33], although all cats were able to meet their energy requirements by visiting only one site—supporting the hypothesis that in domestic cats, spacing between males is a response to the distribution of breeding females [14].

Moreover, these multi-male/multi-female groups show signs of sociality [35]. Adult individuals defend the territory against intruders belonging to different groups [15], and friendly behaviour has been observed among adult females and among adult males and females of the group [35]. In contrast, tolerance with moderate intra-specific aggressive behaviour, but no affiliative behaviour, was observed among adult males, to allow group living [32]. A linear dominance hierarchy, based on the outcome of agonistic interactions (behaviours that manage conflictual relationships between individuals: threat, aggression, submission), develops between males and, independently, between females [32,36,37,38]. When the group of cats is well-established, both males and females live ‘in harmony’ if each cat respects its place in the hierarchy; namely, its own rank, which is a measure of dominance [28]. To speak of rank has meaning only when referring to a group of individuals that stay together long enough to allow a dominance hierarchy to emerge.

Dominant social status is maintained primarily through a set of ritualised cues, rather than through aggressive interactions or even fights [39]. When encountering dominant cats, subordinate cats engage in behaviours such as looking away, lowering their ears slightly, or turning their head away. If the dominant cat walks the same path as the subordinate, the subordinate will deviate from the path to let the dominant cat pass. Dominant cats signal their status through other signals. When approaching a subordinate cat, they will fix their eyes on it, stiffen their forelimbs and hind limbs, raise and rotate their ears so that the opening is lateral, and raise the base of the tail while leaving the rest of the tail low—the tail is comma-shaped [40].

To reinforce their dominance status within their group, dominant cats very often circle their home range, making themselves visible to all cats, maintaining their home range size throughout the year [33]. However, socially dominant cats do not have priority access to food over adult females or kittens until they are one year old [36]. Interestingly, when the dominance hierarchy was measured in different contexts (in the absence of resources and in the presence of food), all adult females of the social group gained in rank [36], and the same occurred in cat colonies where all cats were neutered [37].

The dominant males mark the territory with urine sprayed throughout the year, more than subordinate individuals [37]; the higher the rank, the more frequent the territorial marking behaviour of urine spraying [13]. This is a ‘testosterone-dependent’ manifestation; in fact, it decreases dramatically, and in some individual ceases altogether, following castration [37].

Within the colony, cats show frequent friendly (affiliative) interactions, including greeting ceremonies (nose–nose contact with tails held high), reciprocal rubbing, allogrooming, and passive contact [35,37,38], behavioural patterns reported also at dyadic level [38] (Figure 3).

Cats also show olfactory recognition of group members over strangers [41], while females cooperate extensively in rearing their offspring [15].

Finally, cats recognise colony members and non-members. Most or all colony members show aggression towards unfamiliar cats. Thus, as is the case for most social species, non-members of the group are not allowed to approach and enter the group [13,15].

In summary, domestic cats can live in groups that, under certain conditions, have true organisation which is one of the characteristics of social groups. 

## 4. The Mating System of Unowned Domestic Cats

In rural cats attached to human dwellings and in unowned domestic cats living in the urban environment, females have between one and two oestrus (ovulation) periods per year, averaging four to five days [27,32], with the number of oestrus periods depending on environmental conditions and food resources. Thus, in the Kerguelen archipelago, where the favourable climatic season is very short (a few months), females will only have one oestrus period.

As the cat is an induced ovulator, repeated coitus is necessary for ovulation to occur in most females [42], and ovulation occurs 24–50 h after copulation. Moreover, cat sperm require 24 h to capacitate [43]. Thus, they need to stay within the genital tracts of the female for about 24 h before they can fertilise the eggs. These characteristics of both sexes favour multiple copulations to ensure the fertilisation of all the eggs and, depending on the spatial and social organisation of populations, in some cases favour the mixing of sperm from different males that, in turn, can lead to multiple paternity (i.e., one litter sired by more than one father) [44,45].

It has been observed that the mating system changes between low-density islands and rural environments, as well as high-density urban unowned domestic cat populations. Males, whether social or solitary, compete for access to receptive females regardless of the social organisation of the females.

In the Kerguelen archipelago, no multiple paternity has been found [46], suggesting that the potential for sperm competition is very low there. The lack of behavioural data means it is impossible to determine whether the monopolisation of entire litters by females living in the territory of males is due to a system of polygyny without extra-pair copulations or to monogamy (Figure 4).

In rural areas, behavioural observations [26] have shown that a few males are able to control access to receptive females and perform essentially all copulations, defining a polygynous mating system. The outcomes of fights in males are largely influenced by their phenotypical characteristics [26]: age, experience, physical condition, and body size are all important variables in determining the behavioural tactics adopted by a cat in each encounter [26,47]. Generally, heavier males overpower lighter males in agonistic encounters during the oestrus period of females and have a higher mating success. One example is a male with an all-white coat, homozygous for the W (for White) allele—a dominant allele over any other coat-colour allele [48]. Near the house where this male lived, there were three other houses and two farms with a total of five males and eleven females, none carrying the W allele. This white male, monitored between 1990 and 1992 [49], in 1991 sired 63 (all white) out of 66 kittens delivered by 10 of the 11 females in 18 litters—clearly demonstrating that he had successfully monopolised these 10 females during the breeding season. This male sired no kittens in 1990 and only six kittens in 1992, suggesting that the costs of monopolising females may limit the ability of males to keep competitors out for several consecutive years.

Behavioural observations showed that other males may copulate with females in the absence of the territorial male [26], as the three kittens not sired by the white male suggest. In another rural cat population, Barisey-la-Côte (eastern France), 22% of litters with more than one sire, as well as litters of kittens sired by ‘stranger’ males from outside the population, were identified [44].

In large urban multi-male/multi-female social groups of domestic cats, males and females mate with several partners [16,32,44,47], adopting a promiscuous mating system. Males are not very aggressive during the breeding season, which allows subordinate males to remain in the social group and reproduce [32]. It has been shown [16] that males’ breeding decisions are flexible and depend on the degree of synchronisation of female oestrus; males must choose between defending each courted receptive female until the end of her oestrus period (which can last several days) or leaving her to find a new receptive female. Oestrus of females in colonies usually occurs synchronously [47], but female oestrous may become desynchronised in relation to environmental factors (such as poor weather conditions or prolonged absence of feeders). When many females enter oestrus simultaneously, a single male—even at the top of the hierarchy—can no longer monopolise each female, with males fighting to monopolise females therefore losing breeding opportunities. Rather than defending the females they have just mated with, males adopt a search strategy to gain access to as many females as possible [16,33], regardless of their social rank or body weight [50]—in contrast to cats in rural environments [44]. They spend less time with each female, which considerably decreases the probability of being the father of the kittens, leading to a high rate of multiple paternity. In an unowned domestic cat population living in the neighbourhood of a hospital in the city of Lyon, it has been shown [44] that the proportion of litters sired by at least two males was as high as 80%; in some litters, the numbers of the kittens coincided with the number of fathers.

Conversely, when the oestrus of females is asynchronous, the number of males per receptive female is greater, which increases the competition between them to mate with females, and only the most competitive males (at the top of the social hierarchy) have the highest reproductive success—as occurs in low density rural populations. These same males were unable to monopolise females when their oestrous cycles were synchronous [16]. The variance in male reproductive success was four times greater in years when females bred asynchronously, and dominant males sired most kittens produced. Although the percentage of multiple paternity was the same between the two situations, the proportion of litters where the top males sired at least one kitten was much higher when the females were asynchronous [16,45]. However, no males, even of the highest rank, were able to monopolise an entire litter [16,45].

## 5. Infanticide in Unowned Domestic Cats

Rare cases of infanticide have been observed in rural areas only, a strategy that may have developed to increase reproductive success of wandering males [51]. The method of killing kittens is generally the same as that described for lions [52]: the male bites the kittens on the neck, holding them while shaking them vigorously. All females observed by the authors (D.P. & E.N. [51]) reacted aggressively but were unable to prevent infanticide. One female had two litters attacked where the same male killed some of her kittens. On the second occasion, the female defended the litter in cooperation with the resident male (three years old). The couple managed to deter the attacking male after he had killed the first kitten, and both the resident male and the female showed injuries. No cases of infanticide were recorded in high-density urban colonies where the opportunities appear to be greater, but also where collective defence of litters by females could plausibly be a sufficient deterrent against male attacks. However, this divergence may also be explained by paternity uncertainty in large colonies, where most females mate with several males [53].

## 6. Domestic Cat Behavioural Plasticity as a Gatekeeper of Evolution

The relationship between behavioural plasticity and natural selection has attracted the attention of scientists for a long time [54]. On the one hand, behavioural plasticity allows individuals to respond to ecological pressures by modifying their interaction with new environmental conditions (e.g., exploiting a new food resource) without modifying the genetic diversity of the population. On the other hand, it slows down or even prevents the action of natural selection [55,56,57].

Zuk and collaborators [58,59] suggested that it can be the other way round—that behavioural plasticity might have the greatest influence in supporting new traits.

The intensity of selection for behavioural plasticity depends on the balance between a positive effect it may confer, and a negative one, such as the cost of plasticity on fitness. This balance can lead to different levels of genetic diversity within an adapted population.

The case of the unowned free-ranging domestic cat is particularly interesting because it is possible to compare the characteristics of different populations at an intra-specific level, and allows testing of hypotheses related to the effects of behavioural plasticity and natural selection on evolution of novel traits. There is a large array of traits that can be influenced by behavioural changes: morphological, physiological, and immunological. In fact, all levels of organisation within an organism are integrated by behavioural traits [60,61]. An individual who exhibits behaviour that differs from the original one must change other traits to cope with new abiotic, biotic, or social environments [58]. The latter point is of particular interest in the case of unowned domestic cats in the urban environment, since they face a social environment unknown to ancestral populations.

Therefore, the question we are asking is whether there are indications that new traits (behavioural, morphological, physiological, immunological) are evolving in the domestic cat and what role behavioural plasticity and/or natural selection (allele frequency changes) may have had or have in this process.

## 7. The Evidence That First Triggered Research in This Area

The domestic cat is a dimorphic species: males are, on average, 20% heavier than females [62]. In many polygynous species that show sexual dimorphism, the reproductive success increases with increasing body size [63,64]. But during our first studies in the urban environment [32], what emerged from the observation of sexual behaviour was that high-ranking males, assessed on the basis of the outcome of agonistic encounters, did not copulate more than low-ranking ones. Since copulatory success should, intuitively, correspond to reproductive success, from an evolutionary point of view, this represented an incongruity: the competitive behaviour that leads to facing the costs of being dominant and maintaining the high ranking position (risk of being injured, use of time and energies that could be used in other ways) would be counter-selected and quickly eliminated if it did not have appropriate benefits in genetic terms (high fitness, i.e., a conspicuous transfer of one’s own genes to the next generation). The relationship between social rank and reproductive success has always received considerable attention in the literature. The results of these studies have been controversial and conflicting in mammals: some authors reported a positive correlation between social rank and reproductive success (e.g., [65,66,67,68], whereas others did not [69,70]. It is becoming more evident that dominance rank and reproductive success can be affected by factors other than body size. For example, age (e.g., *Cervus elaphus* L., [71]; *Rattus rattus* L., [72]; *Canis familiaris* L., [73]), prior experience (e.g., *Xiphophorus helleri*, *Heckel 1848*, [74]), or prior possession of resources (e.g., *Sorex araneus* L., [75]), as well as formation of male coalition (e.g., *Gorilla gorilla*, *Savage & Wyman 1847* [76]) can influence the relationship.

Two main hypotheses have been suggested to explain dominance rank and copulatory success relationships in domestic cats. The first concerns a behavioural strategy that high-ranking males might adopt, exploiting certain physiological characteristics of the species’ reproduction, as described above. That is, since a female’s oestrus lasts, on average, 4–5 days, the alpha male and/or high-ranking males may have been selected to take advantage of the concatenation of physiological events, monopolising the most suitable moments.

The second hypothesis concerns the rapid change of the urban environment that some unowned free-ranging domestic cats live in, due to human influence and the consequent abundance of food. Within a very short time, too short in biological terms, the high density and the formation of social groups of cats have occurred, and males may still show the competitive mechanisms of adaptation functional in the original environment, which are no longer effective in the new context. For example, while the dominant male is fighting with another high-ranking male, the lower-ranking males exploit the fact that these two cats are busy fighting and therefore the dominant male cannot prevent them from mating with the female in oestrus, formerly courted by the high-ranking males.

DNA-based paternity analysis studies have shown that the second hypothesis is the most plausible [16,44,45]. The reproductive success of male cats living in both urban and rural environments was compared. Reproductive success was evaluated in terms of the number of kittens sired per year by each male by means of DNA analyses. In rural populations, where females are distributed in small groups around human dwellings, few resident males participate in reproduction, and there is great variability in male reproductive success, with successfully reproducing males being most competitive, and possessing a territory [44]. In urban populations, where males and females live in large groups, on common territories, many resident males participate in reproduction, and there is low variability in male reproductive success [44,45]. The males that do reproduce are not necessarily the most competitive as they exploit the confusion generated by the competitive interactions of the most fearsome males. Molecular results confirm that in the rural, low-density environments, domestic cats are polygynous, as in the original environment of adaptation, while in the urban, high-density environment, they are promiscuous: many males successfully mate with many females at each oestrus. As already mentioned, the percentage of litters generated by at least two males has been found to be 80% in the urban environment, about 22% in the rural environment, and 0% in the sub-Antarctic islands where the cat is feral and lives as a solitary animal [46]. As many as five fathers per litter were found in the urban environment [44].

Furthermore, in support of the second hypothesis, it has been observed that the temporal distribution of oestrus (synchronisation or desynchronisation) represents a variable which can influence the variability of the male reproductive success in a decisive way. If females go into oestrus asynchronously, the situation resembles the ancestral one, with the main difference being that instead of being competitive because a male owns a territory in the rural environment, he is now a high-ranking cat in the urban environment. When oestrus is asynchronous, the age and size of the males is decisive for reproductive success [44,45].

Behavioural and genetic analyses that reported competitive behaviour between adult males and the percentage of kittens they sired showed that unowned urban FRCs ‘make the best of a bad job’. However, this may result in maladaptive behaviour, i.e., performance of risky agonistic behaviour without substantial enhancement of their fitness (Figure 5).

## 8. Further Evidence

Other evidence supporting the hypothesis that, in the new environment, urban FRCs show behaviours that evolved in the original environment of adaptation, is the rank gain of females over males of the same social group when competing for an important resource such as food [36]. In a solitary species, where paternal care is non-existent and where females must be self-sufficient in rearing offspring, it is to be expected that they will become more aggressive when competing for food. This is not the case, for example, with the domestic dog (*Canis l. familiaris*, L.) in similar situations. Although belonging to a domestic species and living in an urban environment with trophic resources comparable to those of FRCs, dogs do not show the same phenomenon: in a social group, the hierarchies of dominance in the absence of resources and in the presence of food do not change [77]. It is possible to assume that evolving from an ancestral social species (*Canis lupus*, L.) makes the difference.

We can interpret the tolerance of adult males towards kittens encountered within the territorial boundaries of a social group of urban cats in a similar way. Since, in the original environment of adaptation, resident males monopolise access to oestrous females living within the boundaries of their territory [13], they have been selected to tolerate all kittens they encounter within it because there is a high probability that they are their offspring. In addition, male tolerance in social group living may be a consequence of the high level of paternal uncertainty resulting from the promiscuous mating system [44,45].

Sexual behaviour of adult females might contribute to this view, although the studies conducted have provided conflicting results. It was found [78] that females may show preferences for unrelated (non-kin) males, and sometimes leave their colony briefly to mate with males from another colony [79]. However, in the urban environment, adult females in oestrus do not show mate choice and mate with all courting males, even strangers [50]. In the original environment, where they lived in individual territories enclosed within the boundaries of a larger male territory, the choice of the resident male occurred before the mating season. The females chose the resources available in that territory, defended also by the resident male, and were mainly monopolised by that male. They have not been selected to exert a mate choice based on morphological or behavioural characteristics (rank).

Finally, in contrast to what is generally proposed, the cooperation in rearing offspring of adult females in urban social groups does not support the social nature of the domestic cat and may be evidence of descent from a solitary species. In a species where there is the typical spatial distribution of many mammals as described above, over large areas, adult females have not been selected to discriminate their own kittens from those of other females. This happens in many other species that live in environments where contact with the offspring of foreign conspecifics is unlikely [80]. If female cats encounter kittens in their territory, there is a high probability that they are theirs and, consequently, they do not discriminate and adopt them.

## 9. Other Interesting Evidence: The Case of the Orange Allele in the Domestic Cat

In cats, the gene responsible for the orange colour is carried on the X sex chromosome. Male cats, which have only one X chromosome, can be orange or another colour (e.g., black, ash grey, brown) depending on the other colouring alleles they carry, but not both. Females, on the other hand, have two X chromosomes and the same individual can have both orange and another colour, a phenotype called tortoiseshell, if the two colours are closely mixed, or calico when the colours form patches (Robinson, 1980). The hair colour expressed by each skin cell is determined randomly, since one of the X chromosomes is inactive during embryological development. The colours may also be white-variegated if the cat also carries the white variegation allele S.

Orange-coloured cats may differ from other cats in several respects. In France, from thirty domestic cat populations, data were collected on 56–491 cats from each population [62]. Two characteristics can be highlighted. Firstly, in rural areas, the frequency of the orange allele is much higher (15–30%, depending on the population) than in urban cat colonies (where there are no or few orange cats). Orange cats show greater sexual dimorphism, with orange males being 32% heavier than orange females, while weight dimorphism is only 16% for other colours—a pattern also found in feral cat populations in Australia [81]. Secondly, orange male cats were significantly more infected with FIV and tended to be less infected with feline leukaemia virus (FeLV) than other males [62]. Such a pattern of infection is consistent with greater aggressive behaviour in orange males, since FIV is transmitted almost exclusively through aggressive male-to-male contact, whereas FeLV is transmitted mainly through social contact [34]. Moreover, orange males were already infected under the age of two years. The difference in age at first infection between the genotypes suggests that orange males interact with other males earlier than non-orange cats.

Thus, due to physical and behavioural differences, orange males may be less flexible in their reproductive strategy. Because they are heavier and more aggressive, orange males may acquire resident/dominant status earlier than other males and achieve greater reproductive success in rural areas where females typically mate with a single male. But in urban cat populations, their aggressive attitude towards other males might be counter-selected. In dense (highly populated) colonies, where females mate with many males, the reproductive success of males may depend more on other behavioural and physiological characteristics than on competitive behaviour among males, as already explained. 

As a result, orange males may spend more time fighting to eliminate numerous competitors rather than mating with females, losing reproductive opportunities and increasing their risk of being wounded. Thus, orange males would disappear from the city because they would not be able to tolerate mates or share food and females in a system where survival and reproduction depend on reciprocal tolerance.

Other behavioural differences have been observed between individuals within the same urban population without an orange cat. It has been shown [82] that cats with bold personality traits have an advantage in terms of reproductive success: they produce more kittens. But, on the other hand, these cats are more likely to be infected with FIV [82]. In the domestic cat, natural selection has probably favoured a proactive temperament despite the cost associated with infection, especially since the disease manifests itself and becomes terminal after a few years, when the cat has already reproduced extensively.

## 10. Is Domestic Cat Behavioural Plasticity Preventing the Evolution of New Traits?

Comparing the behaviour of domestic cats in present-day populations, living according to an ‘ancestral’ model, with the behaviour displayed by ‘derived’ populations can give an idea of which behaviours have not changed despite being faced with novelty.

It might be expected that unowned domestic FRCs that have invaded the urban environment, or which are descended from those that have done so, would behave differently than individuals who have remained in rural or sub-Antarctic environments. Indeed, they can exploit new food resources, but they are apparently less able to reproduce efficiently. In a species where there is no female mate choice [50], the morphological characteristics of males are decisive for reproductive success [45,47]. Evidence of this is the high variability in copulatory and reproductive success of males. In the urban environment, there is lower male variability in copulatory and reproductive success unless females are almost completely asynchronous [45].

However, it is possible that novel traits are becoming established.

In one study in the rural environment [44], only one cat younger than three years sired kittens, whereas two cats aged ten months and many twelve-month-old cats (20–27% per year) produced kittens every year in the urban population (mean age (in years) of fathers: 2.98 ± 1.76 (s.d.); 3.89 ± 2.13 (s.d.) in urban and rural environments, respectively). The reason is intuitive: only morphologically and physiologically fully adult cats have enough experience to become competitive to gain and maintain a territory in the rural environment, in which to include the smaller territories of adult females that will be monopolised when in oestrus. In contrast, in the urban context, males can successfully reproduce as soon as they reach sexual maturity (ten months old). Is the lowering of the age of the first reproduction a new trait that is emerging?

Can less-competitive males, in a situation of promiscuity, be considered satellite males? Can it be considered an emerging novel sexual strategy?

Another trait to be investigated in the domestic cat populations concerns sperm competition. The classical theory regarding the relationship between mating system and testes size claims that the latter increases progressively going from monogamy to polygyny to promiscuity. This is due to the increase of sperm competition, i.e., when gametes from different males compete for fertilising ova, because among the different mating systems, more males compete to fertilise the females and, classically, testes size is expected to be larger in males who experience a higher degree of sperm competition [83]. Domestic cats provide a suitable candidate to investigate testes size in relation to sperm competition and testosterone level at the intra-specific level [84]. It could be assumed that the urban environment is the perfect context for the domestic cat to evolve this trait; although, this has not yet been explored exhaustively.

As Zuk and collaborators highlighted [58], if the behaviours associated with the trait function already exist, a new variant finds a ready foothold, and selection can act accordingly.

## 11. Coexistence between Humans and Domestic Cats in the Anthropocene: Finding the Best Compromise between Animal and Human Welfare and the Preservation of Biodiversity

This section will consider what implication our improved understanding of behavioural plasticity in free-ranging unowned domestic cats (spacing patterns, social organization, mating systems) might have on management of these cats to ensure that their welfare as well as preservation of biodiversity is optimized—and that they are not ‘misunderstood’. Domestic cats are the most abundant carnivores with ~600 million world-wide [85,86]. The presence of free-ranging domestic cats is increasing in all urban and rural environments (e.g., 30–80 million unowned domestic cats in the United States alone [87]), as well as in wild environments on islands (e.g., ~10,000 feral cats on the main island of Kerguelen archipelago [88]). They played an essential role in pest control in past societies and continue to serve this purpose in present rural communities [89]. Globally, cats have a rich and complex relationship with humans of different cultures, but where domestic cats are an invasive species, they negatively impact biodiversity through predation, competition, disease, and hybridisation, and have been implicated in the extinction of reptile, mammal, and bird species [86,90,91]. Notably, domestic cats outnumber wildcats (*F. s. silvestris*), threatening wildcats through hybridisation—although the levels of hybridisation vary considerably between localities [92,93] (for the African wildcat, *Felis. s. lybica*
*Forster 1780*, see [18]). Proposals to prevent such hybridisation include campaigns to neuter domestic cats living near farms or close to forests, with female domestic cats as the primary target since they mate more readily with male wildcats than male domestic cats with wild females [25,92]. In some places, such as Australia and on many small islands where naïve prey species are particularly vulnerable to novel predators, cats are an important threat to some species because of depredation and disease transmission [86,94,95].

This does not seem to be the case in Italy [96], but further studies are needed there and in other countries similar to Italy, where prey species presumably are not particularly vulnerable to predators such as the domestic cat; these countries are characterised by the presence of the domestic cat for a sufficient period of time to have allowed it to co-evolve with its prey in a prey–predator system; in other words, the countries where the domestic cat represents an important ecological component [97].

In Australia, where unowned ‘feral’ cats are killed as part of environmental pest management across the country (urban, rural, and remote areas [98]), an estimated 316,000 cats were killed in 2017–2018 [90]. Yet an estimated 60% of people in Australia own a pet cat [99] and are managed under companion animal ownership legislation and policy [98].

The current geological epoch, characterised by human-induced environmental changes, including large-scale industrial food production, overconsumption of resources, plastic and chemical pollution, climate change, and expanding urban footprint, is commonly referred to as the Anthropocene [100,101]. In countries such as South Africa, pet cats are sometimes killed by caracals (*Caracal caracal*, Schreber 1776), highlighting the fact that urban areas are now increasingly overlapping with home ranges of wildlife [102]. The role of urban cats and their relationship to humans is also changing, particularly in Western countries, where there is growing tension between stakeholders and their views on management and welfare of free-ranging unowned cats, free-ranging (outdoor) pet cats, and confined (indoor) pet cats [101]. People often hold negative views about free-ranging unowned cats who are relatively independent of human care [101], whereas positive but very anthropomorphic views may be held of pet cats who have little independence and may play the role of best friends or even children in a home [103]. Differences in community attitudes towards, and environmental conditions experienced by, unowned free-ranging cats means that a ‘one-size-fits-all’ approach to management and welfare is unlikely to succeed in terms of conflict-free coexistence with humans, conspecifics, or other species [104].

There is growing conflict between stakeholders from different professional groups, including ethologists, behavioural ecologists, veterinarians, wildlife conservationists, animal control municipal authorities, cat welfare professionals, and others, about how to manage and control urban cats, especially when invasive or lethal control methods are used [105]. Stakeholder views may differ in terms of moral, ethical, and value judgements and this influences their views about managing free-ranging cats, with wildlife conservationists more likely to view them as invasive species, and welfare professionals considering them to be ‘homeless’ pets [105]. There is a third position between the two: behavioural ecologists and ethologists regard them as animals still capable of experiencing a non-pet dimension, still under the action of natural selection, with a dignity of their own as domestic animals that is not necessarily given by the place on the sofa at home.

Perhaps the most complex welfare issues and controversial views arise in management of colonies of unowned free-ranging cats, ‘across-the-board neutering’ of cats, and meeting the psychobiological needs of confined pet cats [101], perhaps clearer after the detailed description of the natural behaviour of the domestic cats in this paper.

What is the best solution for maintaining optimal welfare of free-ranging domestic cats while preserving biodiversity and protecting wildcats? Just as for all species, to be able to ensure the welfare of free-ranging domestic cats, it is necessary to know the following: (1) the natural history of the species; (2) the behavioural characteristics in the original environment of adaptation; and (3) the variations in behaviour due to environmental changes. The previous sections show that we can no longer base welfare and management decisions for unowned free-ranging cats in urban environments on knowledge about behavioural ecology of solitary cats living and breeding in more natural ‘wild’ environments. The urban free-ranging domestic cats in the Anthropocene have already responded to the rapid changes in their environment, including abundance of food and higher population densities (of conspecifics) by adapting their behaviour (behavioural plasticity) and social organisation to living in complex social groups, especially those that live in colonies. These urban free-ranging cats are now more social, which has been reflected in different breeding patterns, lower infanticide, more frequent affiliative interactions in general, and different spatial groupings.

The scientific literature is now exhaustive in describing their species characteristics and this knowledge should not be ignored in domestic cat management choices. In order to ensure their welfare, for example, constant territorial displacement and the continuous reshuffling of the social group must be avoided; neutering of the entire social group members and not just certain individuals within it would be desirable; contact and forced constraints must be avoided, but this is intuitive for any living organism.

In Rome, from 1988 to 2017, 1878 colonies of unowned free-ranging cats were monitored, with data collected about animal welfare and social costs. Notably, human attitudes in Rome have shifted from control of these colony cats, within constraints of a ‘No-Kill Policy’ (National & Regional laws), to concern for their welfare [31]. The long-term monitoring of cat colonies in Rome indicates that strategies based on involving and educating people, improving cat food quality, and desire for people to connect with cats, have resulted in decreased seroprevalence of toxoplasmosis and increased empathy and engagement from the human community. Unfortunately, the biodiversity impact of these free-ranging unowned colony cats was not monitored or included in the management strategy (e.g., to preserve birds). In the absence of quantitative data on the predatory impact of cats in Italy, relating exclusively to the urban environment, it is difficult to assess which endangered species may be further threatened by the presence of a co-evolved predator. In other parts of the world, environmental conditions, legislation, and policy may be very different.

To protect biodiversity and maximise health and welfare of cats and humans, scientific evidence based on rigorous monitoring of impacts, including potential harm to cats, wildlife, and humans, is needed [91,95]. Several cat management strategies have been proposed to limit the ecological impact of free-ranging cats across environment types, noting that there is a continuum of possible environments (e.g., natural wilderness, rural, urban, inner city), and for pet cats there are various lifestyles (outdoor, indoor/outdoor, indoor-large house and garden, small apartment, single room). The evidence for/against three of the most popular options (>300 m buffer zones around wilderness/conservation areas, outdoor cat enclosures for pet cats, and cat curfews to keep pet cats indoors overnight) has been previously reviewed [91]. It has been concluded that while many pet owners in the UK and elsewhere are installing cat enclosures, most continue to perceive ‘indoor’-only confinement or curfews as being cruel [91]. The same authors [91] suggested that practical policies are more likely to be adopted and therefore mitigate negative impacts of free-ranging domestic cats in urban fringe areas. Examples of two simple and practical strategies are a requirement of a buffer zone around housing developments, and implementation of physical barriers (natural and/or artificial) to prevent cats from entering wilderness/conservation areas. A very interesting suggestion has come from [106]. Confining pet cats or using collars which warn wildlife of their presence (e.g., bells or ‘Birdbesafe’ collars), may be perceived by owners as being unnatural or too restrictive, and these collars are potentially ineffective in reducing wildlife predation by cats. Cecchetti and her colleagues found that over a 12-week period with 355 cats owned by 219 households in southwest England, the most effective interventions in terms of reducing wildlife being captured and brought home (compared to pre-intervention) were provision of high meat protein/grain-free food (36% decrease) and 5–10 min of daily object play with a ‘fishing’ toy and a ‘mouse’ toy (25% decrease). By contrast, provision of puzzle feeders increased wildlife predation. This finding is remarkable, as a simple change in diet and facilitation of play between pet cats and owners using objects/cat toys would appear to be an easily adopted strategy for owners who care about the welfare of their cats and the protection of wildlife.

## 12. Conclusions

We argue that understanding behavioural plasticity and other recently evolved traits of domestic cats may lead to management strategies that maximise health and welfare of cats, wildlife, and humans. However, translation of our findings into effective cat management strategies will need more research and input from others. Development and implementation of successful cat management strategies that preserve wildlife while protecting cat welfare will need increased collaboration between experts from social and biological sciences, professionals, and community stakeholders. As it has been suggested, [97] ‘cats are not exclusively pets or pests’, and it is necessary to develop a collaborative ‘companion animal ecology’ so that professional stakeholders can link both human–animal domestic relations and ecological processes, for both species (ours included). Dissemination of findings to policy makers and legislators will be important if a suite of evidence based on management approaches is to be developed that factor in the welfare of social groups of free-ranging domestic cats, wildlife, and humans.

## Figures and Tables

**Figure 1 animals-12-01717-f001:**
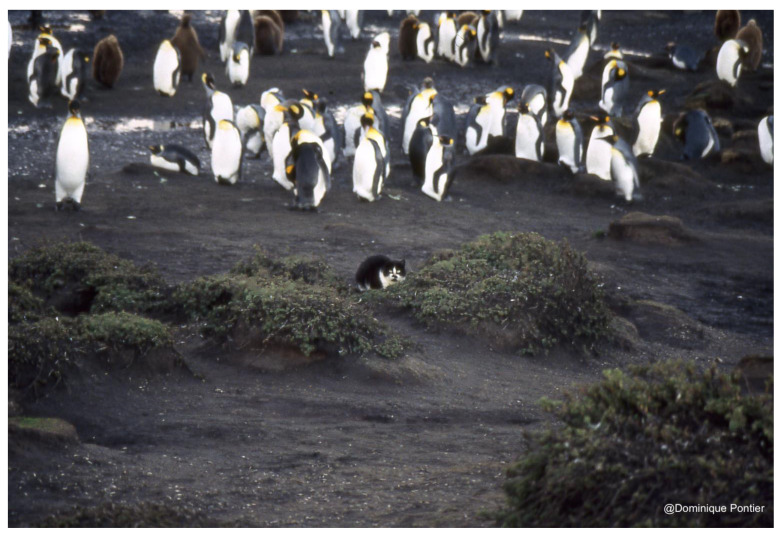
Domestic cat photographed at the Ratmanoff penguin farm zone (Grande Terre, Kerguelen Archipelago). The cats eat the carrion of young penguins or rabbits that have dug burrows in the vicinity of the penguin farm zone (credit Dominique Pontier).

**Figure 2 animals-12-01717-f002:**
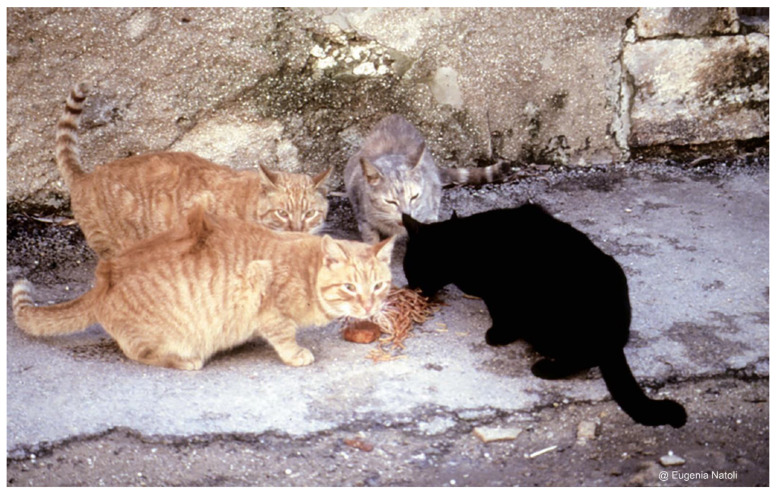
Adult cats living in group in a Sicilian town; they are eating pasta (credit Eugenia Natoli).

**Figure 3 animals-12-01717-f003:**
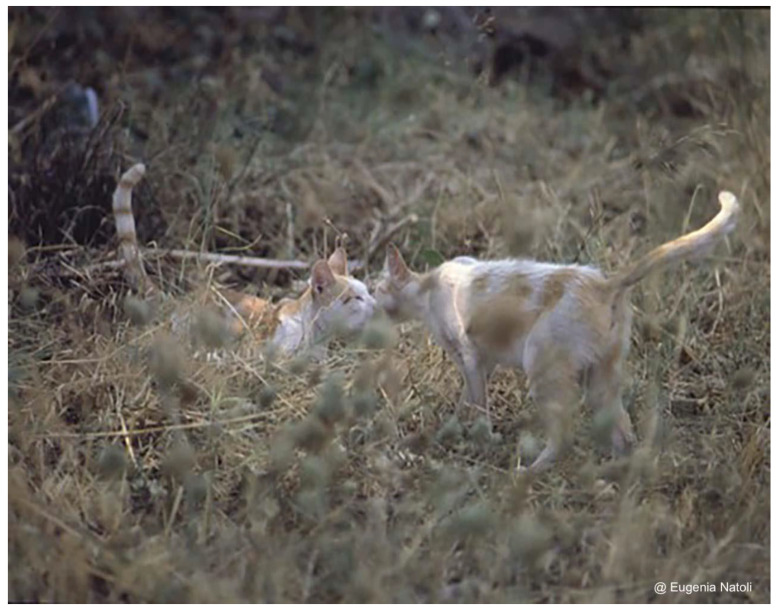
An adult female (on the **right**) and an adult male (on the **left**) belonging to the same social group. They are showing the affiliative behaviour ‘touching nose’ with their tails up [35] (credit Eugenia Natoli).

**Figure 4 animals-12-01717-f004:**
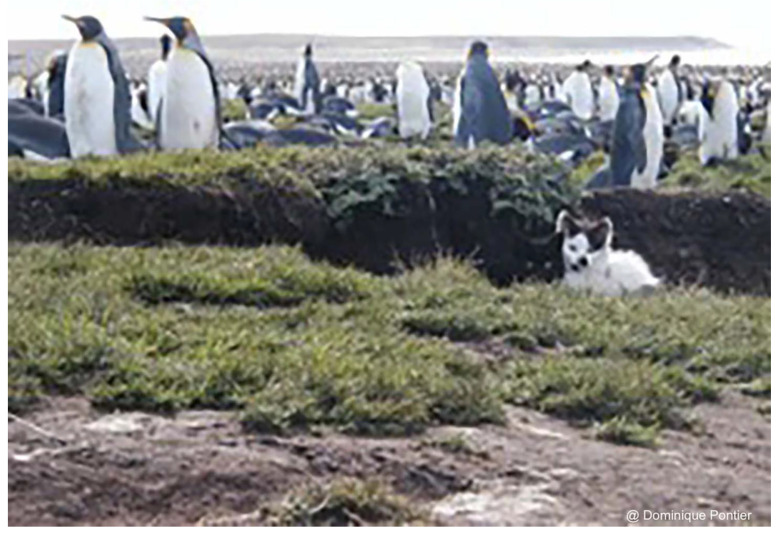
Kittens born at the Ratmanoff penguin farm zone (Grande Terre, Kerguelen Archipelago) (credit Dominique Pontier).

**Figure 5 animals-12-01717-f005:**
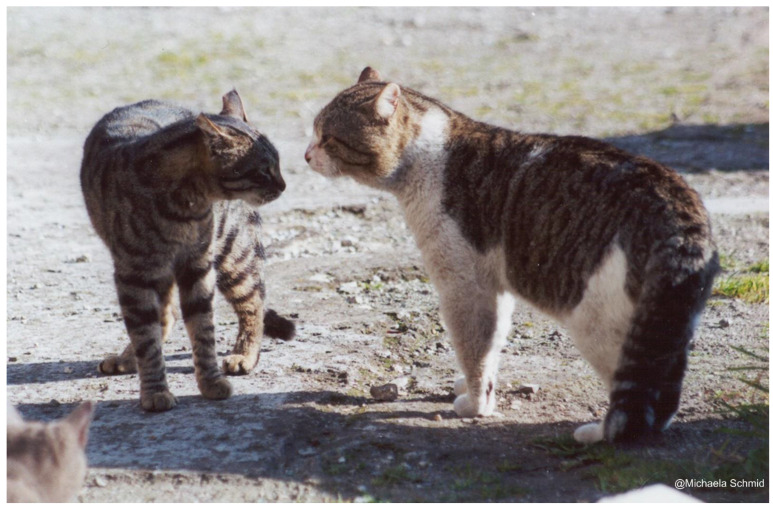
Vocal duel between two males of the same group; the male on the right was the first in the hierarchy, weighed 6 kg, sired 13 kittens, and was positive to feline immunodeficiency virus (FIV); the male on the left was the seventh in the hierarchy, weighed 4 kg, sired 8 kittens, and was FIV negative [45] (credit Michaela Schmid).

## Data Availability

Not applicable.

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
