# Peer review of "Coexistence between Humans and ‘Misunderstood’ Domestic Cats in the Anthropocene: Exploring Behavioural Plasticity as a Gatekeeper of Evolution"

_animals, 2022, doi:10.3390/ani12131717_

Round 1

Reviewer 1 Report

This is an important article linking some aspects of cat behavioural plasticity, ecology and welfare. The smooth coexistence of humans and domestic cats in the Anthropocene surely needs not only improved management techniques, but also adjustments in the theoretical framing of the problem. Nevertheless, to meet the goals from the abstract, authors should pay more attention in the development of their arguments. Also, I think a title should better fit the content of the article: a reader might be confused what “misunderstood domestic cats“ actually refer to. Below are the main points where the article should be revised, but revisions are beneficial throughout the text if the authors wish to increase its comprehension and neatness.

The section 2 would benefit from clearer differentiation among three types of cat ecologies which were mentioned in section 1: solitary animals, urban cats and rural cats: e.g., are lines 98-102 linked only to urban cats?

The authors should be more careful with the use of a notion of the “common ancestor“ (lines 17, 54): the phrase is meaningful only if a predecessor of more than one species is considered. Further, I do not think that “in a rural environment, each cat (or several cats attached to the same farm) has its own feeding location“ (lines 114-115) – they quite often eat at the same time and location, e.g. when they are supplied with milk. The authors should make clear what they mean, considered that feeding routines of farm cats significantly vary among countries and regions.

As far as the main theoretical issues are concerned, I highlight three interconnected problems: (i) which type of cat ecology reminds a putative ancestry condition of Felis s. domestica, (ii) should we consider current specimen of Felis s. silvestris, Felis s. libyca, or Felis s. domestica from the archaeological excavations in the Fertile Crescent (Driscoll et al. 2007, ref. 89; Serpell, J.A. Domestication and History of the Cat. Domest. Cat Biol. Behav. 2000, 2, 180–192) as a proxy for the “natural“ condition of unowned domestic FRCs, (iii) how is behavioural plasticity linked to the putative adaptations functional in the original environment.

Concerning (i), I am not sure about the position of the authors. Based on lines 65-67 and 379-381, I tend to think that the populations consisting of solitary individuals are supposed to meet the ancestry condition. If that is so, I do not think that the sub-Antarctic island environment is a good example, since Felis s. domestica was domesticated in a completely different environment, with much more abundant food resources (the Fertile Crescent). However, the authors may prefer rural unowned cats as a model of the ancestry population. Based on lines 93-97, the hypothesis about the solitary nature of ancestors could be kept. (In lines 431-432, not only rural, but also “mountainous environments“ are mentioned – why?) Based on lines 70-72, one might rather think about the small groups of cats as the ancestry condition (so not solitary, no large groups): the phenomenon of matrilineal social links have been repeatedly observed under different ecological conditions (Turner, D.C. Social Organisation and Behavioural Ecology of Free-Ranging Domestic Cats. Domest. Cat Biol. Behav. 2014, 3, 63–80).

Concerning (ii), I think that data related to the specimen of Felis s. domestica from the archaeological excavations in the Fertile Crescent are the best proxy for putative “natural“ condition of domestic cats (Serpell, J. A., 2014. Domestication and history of the cat. In D. C. Turner, & P. Bateson (Eds.), The domestic cat. The biology of its behavior. Cambridge: Cambridge University Press). The very notion of the Anthropocene reminds us of a blurry line dividing nature and culture: consequently, there is no a priori need to consider a wild species of cat as a resource of the typical behaviours of current domestic cats (as seems to be suggested in lines 65-67; see also "Animal Geographies. Place, Politics, and Identity in the Nature-culture Borderlands", 1998, edited by Wolch and Jody Emel). Rather, “natural“ condition of domestic cats have been formed by several thousands years of the co-evolution with humans, so cats and human societies should be seen as inseparable within the history of the traditional settlements in the “Old World“ (for a notion of “humano-cat“ society, relating to the behavioural plasticity and interconnectedness of both cats and humans, see Jaroš, F. Cats and Human Societies: A World of Interspecific Interaction and Interpretation. Biosemiotics 2016, 9, 287–306). All current individuals of the domestic cat species are derived from the population which was domesticated, so “the original environment of adaptation“ (section 8) is better to be modelled by populations living in some connection to humans, in contrast to truly feral populations (e.g. at the sub-Antarctic island environment). Therefore, I believe that rural cats or current feral populations in the Middle East have a similar ecology as the ancestry populations of Felis s. domestica (food resources, behaviour and sociality).

Problem (iii) concerns a serious theoretical issue. The authors certainly hold the view that cats are highly plastic in their behaviour, probably even more than any other carnivore species. I have argued that such a plasticity even accounts for a term “cat culture“ (Jaroš, F. Cat Cultures and Threefold Modelling of Human-Animal Interactions: On the Example of Estonian Cat Shelters. Biosemiotics 2018, 11, 365–386). Behavioural plasticity enables new behavioural patterns, including some which were likely not exhibited by the ancestry populations (cf. lines 272-274). Thus, I do not think that we should derive the behaviour of current urban populations from “the original environment of adaptation“, as is done repeatedly (e.g. lines 311-314, 349-352, 361-365). 

The article would benefit from neater phrasing of its ideas in some other places. In lines 79-89, an argument is made about the size of territory of cat males and females. 2-3 times larger territory of males is a general feature of Felis s. silvestris, so the sentence in lines 88-89 should not be presented as the outcome of previous considerations. In lines 162-167, the last sentence repeats information from the first one. The argument presented in lines 254-261 is not clear. Behavioural plasticity slows down the action of natural selection, so references 55-57 still contribute to the argument concerning the unmodified genetic diversity of the population. An alternative view (“on the other hand“) should be linked only to references 58-59. The conclusive sentence at lines 345-346 has unclear meaning. The same applies to the sentence at lines 496-497: domestic cats are a significant ecological factor even in the countries without a long co-evolutionary history with other species (e.g. Australia). “Conclusions“, specifically the last two sentences, should be grammatically clearer and better connected to the body of the article.

Eventually, two copy-editing issues need to be fixed:

Pontier should be affiliated to an institution nb. 3.

„Coexistence between humans and domestic cats in the Anthropocene“ and „Conclusions“ need correct chapter numbers.

Reviewer 2 Report

This is an important contribution on cat's behavior. However considreing the impact it may have, it should need a huge clarification of concepts that are loosely or wrongly exposed in a species that lives in a huge variety of conditions. The behavioral plasticity of these species makes itso special and successful. So this needs a much flawless paper

This paper deals with an important issue of a very special species, that of behavioral plasticity.

The authors rightly asserted that taking into account this behavioral plasticity is important to provide efficient welfare measures. They review different aspects of cat lives  namely spacing patterns, intraspecific tolerance, mating systems among others. Despite a well-documented review, the paper contains many contradictions and confusing issues.

Though correctly stating -line 75- that Home Ranges should not be confused with territories, the definitions itself is confusing as it is mentioned that territories are included in home ranges. This is quite unconvetional definition. Generally a territory is an area -home range- that is defended and thus is exclusively used by an individual or a group of individuals. Therefore if a species is territorial one can find a mosaic of non-overlapping area. Individuals from this species may defend this area against same-species intruders in patroling along the borders and/or by proclaiming or marking boundaries. In addition individuals from the same species finding markings or listening acoustical proclamations should avoid this area and should not trespass the boundaries. Many authors have provided evidence that it is not the case and that cats sniffing urine sprays, overmark and continue their way and do not turn back as it should be the case if they respect the « territorial markings ». In addition many observations do not document that intruding cats are only attacks at the boundaries of the territory. The authors of this paper at different places mentioned overlaping home ranges, which is correct but also « overlaping territories »  which is not correct. Therefore this issue is very confusing and should be clarified. Functions of urine spraying, other than marking a teritory, might be evoked and in no case ruled out. Abundant evidences are in line of a non territorial species whatever the conditions of rural free-ranging owned cats or unowned cats. These evidences could not be overlooked.

The second important issue is that of sociality. Throughout the text, the term social is used, and in many cases like in a useless way as a verbal tic. The definition of a social species is a very loose one and ignore most of the literature on social sspecies whether wild or domestic. It should be recalled that sociality if a trait that has evolved in some classes of vertrebrates and invertebrates during the course of evolution. Sociality, as mentioned by many authors, is based on inter-attraction between individuals of the same species. This inter-attraction leads individuals to constitute stable groups that included mating systems and several generations. As pointed out as early as in Aristotle this should not be confused with of course solitary species and also with gregarious species. In these gregarious species, groups are made for ecological reasons no because of intraspecific inter-attraction. Of course in both gregarious and social species there exists a tolerance between individuals. Nowhere in this paper it is stated that groups of cats, that may also live solitary, may group for ecological reasons and that cats may display a full range of inter-individual tolerance from solitray non tolerant individuals to more tolerant ones in colonies (see line 112). Authors such as Liberg very clearly showed that this tolerance is dependent on the abundance and dispersion of food ressources and nowhere inter-attraction wa mentioned. Line 107 there is a good statement of what is dominance. However there is only few loose arguments that allows to consider Felis catus as a social species rather than an oppurtinistic species showing all levels of intraspecific tolerance. Constructing a dominance hierarchy is an exercise that may not have a biological meaning especially when in a trivial way it is based on age and weight. In « true » social species the criteria for dominance are much more complex (cf Bernstein 1982). It should also be recalled that a relationship that may be qualified as social, has TWO components, one widely used – to determine dominance ranks and building a hierarchy,  a negative one based on conflict resolution and one, the affiliative -friendly- component, that leads to close proximity between individuals, and is responsible to group cohesion. Unfortunately this essential positive component is very often overlooked. In this paper it is mentioned that in groups « frequent friendly » interactions are observed -line 154-. However when describing the social organization of a group – in Hinde 's acception, one should describe in each dyad of the group the balance between friendly and agonistic interactions. Once more « social » should not be a verbal tic each time one observed a group of animals. This paper contains a lot of contradictory statements (in dbeate about territoriality and sociality – see lines 357-360 ) or loose arguments about concepts that have received a lot of attention in other species.

To conclude this paper is important by its rationale of comparing the many living condition of this ubiquitous species, espceially in order to provide suitable and efficient welfare. Howver the two important issues -spacing patterns and grouping patterns should to be much more clarified.

Reviewer 3 Report

This is an interesting and well-written review article by several experts in the field of free-ranging cat behavior. I have just a few minor comments that I hope the authors will find helpful.

RC1: In the Introduction, please include the scientific name of the domestic cat (since you list scientific names for other species at L49).

RC2: Page 2 footnote, please clarify your definition of “feral cats”. What specifically do you mean by being returned to a “feral situation”? Does this just mean that they live independent of people (i.e., they catch their own food) and lack exposure to humans (i.e., no human socialization)? How would you classify a FRC that exploits human resources (e.g., garbage and refuse) but lacks human socialization? 

RC3: L79, For your statement, “The domestic cat is a territorial animal, whatever environment it lives in” is this always true? Is it possible that given the behavioral plasticity of cats, territorial behavior may depend on life experience and resource distribution?

For example, research exploring the distribution of scent marks in outdoor cats (kept in a large, fenced-in area) indicated that cats were not using scent marks to define territorial boundaries. The author states that “Cats have rarely been seen to leave or avoid an area after investigating marks, suggesting that the marks serve as information markers rather than as an active method of deterrence”. This author argues the data support the idea that “cats do not defend territories, instead patrolling and reinforcing marks throughout a looser home range” (Feldman, 1994). 

Other research has also showed substantial overlap in cat ranges with little defense, even among males (which sometimes had smaller ranges than females). For example, Mertens & Turner (1986) state in their research of FRC home range size that “The general pattern of social organization found elsewhere was confirmed: males were generally more tolerant of each other than females (based on range overlap), especially considering animals living on different farms. Animals from the same primary home showed considerable range overlap. Male ranges were much smaller than expected (not even twice as large as the female ranges, whereas they should have been about 10 times the size, based on all other studies).”

My point is that although cats may actively defend territories in some instances and males may sometimes have larger territories than females, in my opinion the data have not been consistent to make the statement that this is always true and that these patterns are uniform across all FRCs.

Feldman (1994). Methods of scent marking in the domestic cat. https://doi.org/10.1139/z94-147

Mertens & Turner (1986) Home Range Size, Overlap and Exploitation in Domestic Farm Cats (Felis Catus). https://www.jstor.org/stable/4534554

RC4: L75, The statement “Home range and territory should not be taken as synonyms” is made. However, in the paper the authors seem to continue to use both the term “territory” (L98) and “home range” (L120) when discussing socially living cats. Can you please clarify this? In what instances is “territory” the appropriate usage and in what instances is “home range” appropriate? How it is currently written it seems that in some cases cats do not defend territories and instead hold home ranges, which is counter to your statement at L79.

RC5: L448, “Is the lowering of the age of the first reproduction a new trait that is emerging?” I don’t have an edit here, I just found this statement interesting since earlier and more frequent reproduction is one outcome of the domestication process in general.

RC6: L467, The section on managing FRC to best compromise for coexistence between FRCs and humans is excellent. The only study that isn’t mentioned that would be worth including as a potential recommendation is research that demonstrated that “Providing high-meat-protein food and object play both reduce predation by cats” (Cecchetti et al. 2021).

Martina Cecchetti, Sarah L. Crowley, Cecily E.D. Goodwin, Robbie A. McDonald (2021). Provision of High Meat Content Food and Object Play Reduce Predation of Wild Animals by Domestic Cats. Current Biology. https://doi.org/10.1016/j.cub.2020.12.044.

Round 2

Reviewer 2 Report

I appreciate very much the efforts the authors made in reviewing their text.

Event though I also respectfully disagree with some arguments, I enjoy the debate and wish it remains instead of giving the impression of being settled.

The authors are perfectly right when they said that they use a wide set of available literature to base their arguments and that even if I still disagree with them, I should accept them. This is what I am going to do.

I have an extensive expertise on social behavior, social relationships, socialization, on other species which I used in my own observations on cats – so it is not an « opinion » but the question of how these concpets are relevant to decribe the functioning of group of cats une the conditions where they are formed – which is not « universal » as it is for social species. However I should admit that my expertise on cat groups is much less extensive that the expertise of the authors. And there is especially an important point that it is mentioned in the cover letter : that their observations on cat groups in Rome are on unneutered individuals. If not already higlighted I think this should be emphasized.

When in the cover letter, the authors said that « they can live in groups that have a truly social organization under certain conditions. », they illustrate excatly the point I am very familiar with about the use of the term « social ». One could have written « they can live in groups that, under certain conditions, have a truly organization which is one of the characteristics of social groups », I would have more appreciate this.

I am quite sure that some organization can be observed in groups of gregarious species that are not social.

The term gregarious is totally absent in this text – If my reading is OK!

Though once more the expertise of the authors and the extensive literature they used is impressive, it still lacks some papers from Leyhausen and his collaborators that could have fed the debate.

Again I would appreciate if this debate remains open and constantly fed with data even controversial as the cat is, for me - »in my opinion » !, a species which keeps ethologists' eyes opened by constantly challenging the concepts.

Again many respectful thanks to the authors for their paper.
